

# Single fraction stereotactic radiosurgery and fractionated stereotactic radiotherapy provide equal prognosis with overall survival in patients with brain metastases at diagnosis without surgery at primary site

Garett Paul Ostdiek-Wille[1,*], Saber Amin[2,*], Shuo Wang[2], Chi Zhang[2] and Chi Lin[2]

[1] College of Medicine, University of Nebraska Medical Center, Omaha, NE, United States of America
[2] Radiation Oncology, University of Nebraska Medical Center, Omaha, NE, United States of America
* These authors contributed equally to this work.

Corresponding author
Chi Lin, clin@unmc.edu

## ABSTRACT

**Background and purpose.** Stereotactic radiosurgery (SRS) and fractionated stereotactic radiation therapy (SRT) are both treatments shown to be effective in treating brain metastases (BMs). However, it is unknown how these treatments compare in effectiveness and safety in cancer patients with BMs regardless of the primary cancer. The main objective of this study is to investigate the SRS and SRT treatments' associations with the overall survival (OS) of patients diagnosed with BMs using the National Cancer Database (NCDB).

**Materials and methods.** Patients in the NCDB with breast cancer, non-small cell lung cancer, small cell lung cancer, other lung cancers, melanoma, colorectal cancer, or kidney cancer who had BMs at the time of their primary cancer diagnosis and received either SRS or SRT as treatment for their BMs were included in the study. We analyzed OS with a Cox proportional hazard analysis that adjusted variables associated with improved OS during univariable analysis.

**Results.** Of the total 6,961 patients that fit the criteria for the study, 5,423 (77.9%) received SRS and 1,538 (22.1%) received SRT. Patients who received SRS treatment had a median survival time of 10.9 (95% CI [10.5–11.3]), and those who received SRT treatment had a median survival time of 11.3 (95% CI [10.4–12.3]) months. This difference was not found to be significant (Log-rank $P = 0.31$). Multivariable Cox proportional hazard analysis did not yield a significant difference between the treatments' associations with OS (Hazard Ratio: 0.942, CI 95% [0.882–1.006]; $P = .08$) or SRS *vs.* SRT.

**Conclusions.** In this analysis, SRS and SRT did not show a significant difference in their associations with OS. Future studies investigating the neurotoxicity risks of SRS as compared to SRT are warranted.

## BACKGROUND

Most malignant tumors within the brain are metastases from primary cancer elsewhere in the body (*Gaspar et al., 1997*). It has been estimated that 8.5% to 9.6% of all cancer patients develop brain metastases (BMs), but the actual occurrence rate is likely higher according to autopsy studies (*Sacks & Rahman, 2020*) and varies based on the type of primary tumor. This is evident in the findings that BM development occurs in 40% to 50% of lung cancer patients, 5% to 15% of breast cancer patients, around 10% of melanoma patients, 0.6% to 3.2% of CRC patients, and between 4% and 17% of advanced renal cancer patients (*Christensen et al., 2016*; *Sheehan et al., 2003*). The site of the primary cancer can impact the median overall survival (OS) with all falling between 4 and 16 months (*Michl et al., 2015*; *Fabi et al., 2011*; *Esmaeilzadeh et al., 2014*).

Current treatments for BM include surgical resection, whole brain radiation treatment (WBRT), single fraction stereotactic radiosurgery (SRS), fractionated stereotactic radiation therapy (SRT), systemic steroids, best supportive care as well as combinations of these treatments (*Arita et al., 2014*). Factors influencing the choice of treatment modality for patients with BM are number and size of BMs, histology, and the status of extracranial disease control. Patients with many BMs, especially less than 10 lesions are more likely to receive SRS/SRT (*Yamamoto et al., 2014*), while patients with a metastasis greater than four centimeters in size would more likely receive surgery with or without SRT to the surgical bed. For patients with solitary BM, surgical resection can improve survival outcomes (*D'Andrea et al., 2017*). For others, surgical resection is utilized to help improve neurological functioning and mass reduction (*Arita et al., 2014*). However, surgical resection is rarely used without combining it with radiation treatment due to high rates of reoccurrence (*Amin et al., 2020b*). WBRT has been the traditional approach to treat BM; however, more recent studies have revealed that targeted treatments such as SRS and SRT are potentially more beneficial clinically, as well as present less risk for radiation toxicity (*Arita et al., 2014*; *Gu et al., 2019*). SRS and SRT can provide high doses of radiation to a specific area rather than exposing the entire brain to radiation (*Jimenez et al., 2017*).

SRS and SRT vary in dose per fraction and the number of fractions utilized. SRS utilizes a single fraction of high dose radiation (*Shaw et al., 1996*) but is associated with certain toxicities such as radiation-induced brain necrosis (RN) or intralesional hemorrhage (*Jimenez et al., 2017*). SRT is a stereotactic radiation treatment in which the total dose is divided into three to five fractions, delivered on separate days of treatment. SRT could potentially help reduce SRS-associated toxicities especially in specific areas of the brain that exhibit a higher sensitivity to radiation. A study reported that the visual pathway (chiasm and optic nerves) is more sensitive to radiation than other cranial nerves (*Leber et al., 1995*). SRT could be more appropriate than SRS to be used to treat tumors compressing or invading the visual pathway.

Toxicity from radiation is a concern when considering either SRS or SRT as a treatment plan. Perhaps the most concerning adverse outcome of these two radiation treatments is RN. Previous research has found that about 9% of patients receiving SRS as the primary form of radiation treatment for BM's develop RN at one year with the figure doubling by year two following treatment. RN develops between 2- and 32-months post-treatment and patients can experience seizures as well as speech, motor, and cognitive deficits (*Minniti et al., 2011*). Having a larger tumor targeted for treatment has been significantly linked with a higher likelihood of developing RN (*Donovan, Parpia & Greenspoon, 2019*), but evidence of SRT reducing the risk has been inconclusive and whether reduction in RN can be translated to OS benefit has been inconclusive (*Donovan, Parpia & Greenspoon, 2019*; *Minniti et al., 2014*).

Therefore, the objectives of the current study are to use the National Cancer Database (NCDB) to investigate the factors that are associated with receiving SRT treatment as compared to SRS treatment and to evaluate if there is any difference between the association of SRT and that of SRS with the overall survival of cancer patients with BMs.

## MATERIAL AND METHODS

### Data source

This study utilized data from the National Cancer Database (NCDB). The NCDB is a venture of both the American College of Surgeons and the American Cancer society. The database is provided with information from hospital registries from Commission on Cancer accredited facilities. More than 70% of newly diagnosed cases of malignant cancer are accounted in the NCDB. Since the data extracted for the study was previously de-identified, there was not a need for Institutional Review Board (IRB) approval.

### Study population

The population for this study was comprised of individuals who were 18 years of age or older who had BMs at the time of diagnosis of breast cancer, non-small cell lung cancer (NSCLC), small cell lung cancer (SCLC), other lung cancers, melanoma, colorectal cancer (CRC), or kidney cancer and had received either SRS (dose of 1,500–2,400 Gy (1 fraction)) or SRT (dose of 2,100–3,000 Gy (three fractions), and dose of 2,500–3,250 Gy (5 fractions)) as treatment for their BMs. Patients with diagnoses between 2010 and 2015 in the NCDB were considered for the study. The year 2010 is the year when the NCDB first started collecting information about BMs at the time of the diagnosis of primary cancer. Individuals with surgery at the primary cancer site were excluded from the study because our previous study has shown that these patients belong to a different cohort with a much higher overall survival (*Amin et al., 2020a*). Information about brain surgery is not reported in the database.

Biological Effective Dose (*BED) calculation*
$BED_{10}$ of 24 Gy in 1 fraction is 81.6 Gy
$BED_{10}$ of 18 Gy in 1 fraction is 50.4 Gy
$BED_{10}$ of 15 Gy in 1 fraction is 37.5 Gy
$BED_{10}$ of 27 Gy in 3 fraction is 51.3 Gy

BED$_{10}$ of 30 Gy in 5 fraction is 48 Gy

### End points

Overall survival (OS) was the primary outcome for the study. OS was calculated from the date of diagnosis until the date of the individual's death. Those alive or lost to follow-up were censored. Predictors of receiving SRS rather than SRT were also included in the study. These predictors were reported using the odds ratio (OR).

### Predictors or explanatory variables

OS was evaluated and analyzed between the groups receiving SRT as compared to SRS. Other variables of interest were age at diagnosis were age at diagnosis, sex, race, education level, income, place of living, hospital type, insurance status, Charlson/Deyo score, chemotherapy treatment, the type of primary cancer, and the year of the initial diagnosis. The predictors for whether SRS or SRT treatment was received include sex, race, education, income, place of living, hospital type, insurance status, Charlson/Deyo score, chemotherapy treatment, immunotherapy treatment, primary cancer type, and the year of primary cancer diagnosis.

### Statistical analyses

Descriptive statistics for categorical and continuous variables are reported. Predictors of SRS use as compared to SRT were identified using the logistic regression model. The OR was reported as the measure of association with likelihood of using SRS. Survival time was measured in months from the date of diagnosis to the date of death. We used the Kaplan–Meier (KM) method to generate survival curves and analyzed the differences between groups using the log-rank test. Cox proportional hazards regression analysis was conducted to estimate the hazard ratio (HR). Univariable Cox proportional hazard ratios were first obtained, and multivariable Cox proportional hazard ratios were determined including variables that had a significance of $P < .15$ in the univariable analysis. Analyses were conducted using SAS 9.4 (SAS Institute Inc.).

## RESULTS

### Patient's characteristics

The study's population had a total of 6,961 participants with 3,519 (50.6%) being male. All participants fell between the ages of 21 and 90 with a median age of 65.0 years old. Among all patients, 5,423 (77.9%) received SRS while 1,538 (22.1%) received SRT. The median (range) of ages in patients receiving SRS treatment was 65 (21–90) years old, and the median (range) of ages in patients receiving SRT was 65 (24–90) years old. Of the 6,961 participants, 5,887 (85.4%) were white, 743 (10.8%) were black, and 267 (3.9%) belonged to other races. The majority of participants had an income of greater than $35,000 per year (61.3%), lived in an urban setting (98.3%), had health insurance (97.0%), had a Charlson/Deyo score of 0 (67.0%), and had received chemotherapy (72.2%) but not immunotherapy (92.4%). Of the primary cancer types, 171 (2.5%) had breast cancer, 5,595 (80.4%) had NSCLC, 204 (2.9%) has SCLC, 222 (3.2%) has other types of lung cancer, 449 (6.5%) had melanoma, 67 (1.0%) had CRC, and 253 (3.6%) had kidney cancer. All of the study's demographic data is presented in Table 1.

**Table 1 Baseline characteristics of BMs patients who received SRS or SRT.** Shows the frequency and proportion of baseline characteristics by SRS and SRT.

| Variable | | SRS N (%) 5,423 (77.9) | SRT N (%) 1,538 (22.1) | Total 6,961 |
|---|---|---|---|---|
| Age at diagnosis, Median (range) | | 65.0 (21–90) | 65.0 (24–90) | 65.0 (21–90) |
| Sex | Male | 2,733 (50.4) | 786 (51.1) | 3,519 (50.6) |
| | Female | 2,690 (49.6) | 752 (48.9) | 3,442 (49.5) |
| Race | White | 4,580 (85.2) | 1,307 (85.8) | 5,887 (85.4) |
| | Black | 588 (10.9) | 155 (10.2) | 743 (10.8) |
| | Other | 205 (3.8) | 62 (4.1) | 267 (3.9) |
| | Unknown | 50 | 14 | 64 |
| Education NHSD | >=13% | 2,178 (40.2) | 652 (42.5) | 2,830 (40.7) |
| (No High School Degree) | <13% | 3,235 (49.8) | 881 (57.5) | 4,116 (59.3) |
| | Unknown | 10 | 5 | 15 |
| Income | >=$35,000 | 3,310 (61.2) | 944 (61.6) | 4,254 (61.3) |
| | <35,000 | 2,101 (38.8) | 589 (38.4) | 2,690 (38.7) |
| | Unknown | 12 | 5 | 17 |
| Place of Living | Urban | 5,175 (98.1) | 1,477 (98.8) | 6,652 (98.3) |
| | Rural | 98 (1.9) | 18 (1.2) | 116 (1.7) |
| | Unknown | 150 | 43 | 193 |
| Hospital Type | Academic | 2,501 (46.8) | 699 (46.1) | 3,200 (46.7) |
| | Community | 2,843 (53.2) | 817 (53.9) | 3,660 (53.4) |
| | Unknown | 79 | 22 | 101 |
| Insurance Status | Yes | 5,197 (96.9) | 1,477 (97.2) | 6,674 (97.0) |
| | No | 166 (3.1) | 43 (2.8) | 209 (3.0) |
| | Unknown | 60 | 16 | 78 |
| Charlson/Deyo Score | 0 | 3,686 (68.0) | 976 (63.5) | 4,662 (67.0) |
| | 1 | 1,209 (22.3) | 380 (24.7) | 1,589 (22.8) |
| | >=2 | 528 (9.7) | 182 (11.8) | 710 (10.2) |
| Chemotherapy | Yes | 3,979 (73.4) | 1,045 (68.0) | 5,024 (72.2) |
| | No | 1,444 (26.6) | 493 (32.1) | 1,937 (27.8) |
| Immunotherapy | Yes | 406 (7.5) | 123 (8.0) | 529 (7.6) |
| | No | 5,017 (92.5) | 1,415 (92.0) | 6,432 (92.4) |
| Cancer Type | Breast | 127 (2.3) | 44 (2.9) | 171 (2.5) |
| | NSCLC | 4,407 (81.3) | 1,188 (77.2) | 5,595 (80.4) |
| | SCLC | 162 (3.0) | 42 (2.7) | 204 (2.9) |
| | Other Lung | 159 (2.9) | 63 (4.1) | 222 (3.2) |
| | Melanoma | 320 (5.9) | 120 (8.4) | 449 (6.5) |
| | CRC | 45 (0.8) | 22 (1.4) | 67 (1.0) |
| | Renal Cell | 203 (3.7) | 50 (3.3) | 253 (3.6) |
| Year of Diagnosis | 2010–2013 | 2,931 (54.1) | 709 (46.1) | 3,640 (52.3) |
| | 2014–2015 | 2,492 (46.0) | 829 (53.9) | 3,321 (47.7) |

## Factors associated with the use of SRS *vs.* SRT

Using univariable logistic regression analysis to search for factors found to have an association with the type of treatment received (SRS or SRT) yielded several significant results (Table 2). Individuals with either a Charlson/Deyo Score of 1 or 2 or more were significantly more likely to receive SRS treatment than those who have a score of 0. Patients with certain types of primary cancers (*Other lung,* melanoma, and CRC) were significantly more likely to receive SRT treatment when compared to kidney cancer. Other factors significantly linked with receiving SRS treatment as opposed to SRT treatment were receiving chemotherapy treatment and having diagnosis before 2014. Age, sex, race, education level, income, place of living, hospital type, insurance status, other types of primary cancers. (NSCLC and SCLC) were not found to have a significant association with SRS or SRT received.

Multivariable logistic regression analysis was conducted while controlling for the significant factors from the univariable logistic regression analysis. Individuals with a Charlson/Deyo score of 1 or 2 or more, a certain primary cancer (*Other lung,* melanoma, and CRC) were both significantly linked with receiving SRT treatment when compared to kidney cancer. Having a diagnosis before 2014 was significantly linked to receiving SRS treatment. Education level of the area the individual was from, income, place of living, and other primary cancers (NSCLC and SCLC) were not found to have an association with the use of SRS when compared to the use of SRT.

## Survival analysis

Comparing the OS of both the individuals receiving SRS and SRT treatments were analyzed using KM curves (Fig. 1). Individuals receiving SRS treatment had a median survival time of 10.9 (95% CI [10.5–11.3]) months. Meanwhile, individuals receiving SRT treatment had a median survival time of 11.3 (95% CI [10.4–12.3]) months. This difference was not found to be significant (Log rank $P = 0.31$).

## Cox proportional hazard analysis

When analyzing the HR using univariable Cox proportional hazard analysis (Table 3), it was determined that receiving SRS treatment was not associated to significantly different survival outcomes (HR: 0.968, 95% CI [0.908–1.032]; $P = .31$) when compared to receiving SRT treatment. A younger age at diagnosis, being female and an individual belonging to a race that was neither white nor black had a significant association with improved OS. Additional factors with significantly positive association with OS were receiving care from an academic hospital, living in an area with income >$35,000, living in an area with < 13% of people with no high school degree, a lower Charlson/Deyo comorbidity score, prior reception of chemotherapy treatment, certain primary cancers (Breast, NSCLC, SCLC, and melanoma), and a diagnosis in the years 2014–2015 compared to 2010–2013. Meanwhile differences based on race (identifying as black or white), place of living, insurance status, and having other lung cancers (not NSCLC or SCLC) were not found to be associated with OS.

In the multivariable Cox proportional hazard analysis adjusted for factors that were significant in the univariable Cox regression analysis (Table 3), there was no significant

**Table 2** Univariable and Multivariable logistic regression analysis for receiving SRS.

| Variables | | Univariable analysis OR (95% CI) | P-value | Multivariable analysis OR (95% CI) | P-value |
|---|---|---|---|---|---|
| Age | | 1.01 (0.99–1.01) | 0.69 | … | |
| Sex | Male | Ref | | Ref | |
| | Female | 1.03 (0.92–1.15) | 0.62 | … | |
| Race | White | Ref | | Ref | |
| | Black | 1.08 (0.90–1.31) | 0.41 | … | |
| | Other | 0.94 (0.71–1.26) | 0.69 | … | |
| Education | >=13% NHSD | 0.91 (0.81–1.02) | 0.11 | NS | 0.22 |
| | <13% NHSD | Ref | | Ref | |
| Income | >=$35,000 | Ref | | Ref | |
| | <35,000 | 1.02 (0.91–1.14) | 0.77 | 0.88 (0.72–1.08) | 0.23 |
| Place of Living | Urban | Ref | | Ref | |
| | Rural | 1.55 (0.94–2.57) | 0.09 | 1.57 (0.94–2.61) | 0.09 |
| Hospital Type | Academic | Ref | | Ref | |
| | Community | 0.97 (0.87–1.09) | 0.63 | … | |
| Insurance Status | Yes | Ref | | Ref | |
| | No | 1.10 (0.78–1.54) | 0.59 | … | |
| Charlson/Deyo Score | 0 | Ref | | Ref | |
| | 1 | 0.84 (0.74–0.96) | 0.01 | 0.84 (0.73–0.96) | 0.01 |
| | >=2 | 0.768 (0.640–0.922) | 0.01 | 0.78 (0.65–0.94) | 0.01 |
| Chemotherapy | Yes | Ref | | Ref | |
| | No | 0.77 (0.68–0.87) | <0.01 | 0.82 (0.72–0.94) | <0.01 |
| Primary Cancer | Breast | 0.71 (0.45–1.13) | 0.15 | 0.70 (0.44–1.13) | 0.15 |
| | NSCLC | 0.91 (0.67–1.25) | 0.58 | 0.90 (0.65–1.25) | 0.52 |
| | SCLC | 0.95 (0.60–1.50) | 0.83 | 0.95 (0.59–1.53) | 0.84 |
| | Other Lung | 0.62 (0.41–0.95) | 0.03 | 0.63 (0.41–0.97) | 0.04 |
| | Melanoma | 0.61 (0.42–0.89) | 0.01 | 0.65 (0.44–0.96) | 0.03 |
| | CRC | 0.50 (0.28–0.91) | 0.02 | 0.50 (0.27–0.92) | 0.03 |
| | Renal Cell | Ref | | Ref | |
| Year of Diagnosis | 2010–2013 | 1.38 (1.23–1.54) | <0.01 | 1.36 (1.21–1.51) | <0.01 |
| | 2014–2015 | Ref | | Ref | |

**Notes.**

Trt, treatment treatment; NHSD, no high school degree.

difference in OS for individuals who received SRS treatment compared to SRT (HR: 0.942, CI 95% [0.882–1.006]; $P = .08$). However, female sex, non-white race, younger age at diagnosis, high income-level, receiving treatment at an academic center, and having a primary cancer that is either breast cancer, NSCLC or melanoma, a Charlson/Deyo comorbidity score of 0, receiving chemotherapy, and having a diagnosis date after 2014 were all associated with improved OS.

Further subset-analyses were conducted to determine the associations of SRT *vs.* SRS with OS in patients with different primary cancer sites. These results (Table 4) showed a

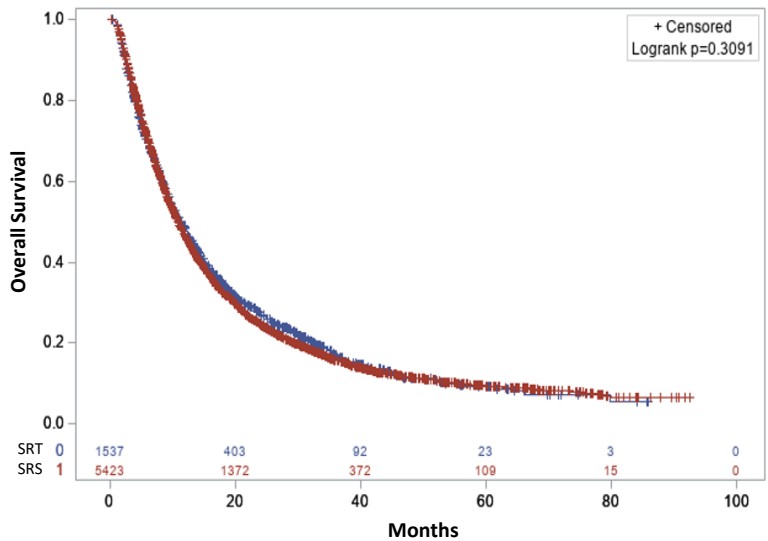

**Figure 1 Overall survival.** Stereotactic radiosurgery (Single fraction SRS: color red) and stereotactic radiotherapy (3-5 fractions SRT: color blue).

significant improved OS for patients with CRC receiving SRT treatment compared to SRS (HR: 0.417, CI 95% [0.204–0.854]; $P = .01$).

Biological Effective Dose (BED) is an important factor to consider when comparing survival outcomes. For this reason, the association of SRS *vs* SRT with OS was also analyzed after stratifying SRS and SRT by $BED_{10}$ of 55 Gy. Additionally, this analysis was further conducted in each primary cancer site (Tables 5 and 6). Univariable and further multivariable Cox proportional analysis revealed no significant differences in OS when comparing patients receiving either SRS or SRT and either above or below a $BED_{10}$ of 55.

# DISCUSSION

The goal of the study was to analyze the differences in OS between patients who received SRS to treat BMs and those who received SRT. Our analysis did not reveal a significant association between the type of RT and the improved OS in either univariable or multivariable Cox regression analysis.

The median survival (11.3 months) found in our study for patients receiving high dose radiation treatment over several fractions (SRT) was slightly lower than a reported survival from a single institution's data (14.8 months) (*Minniti et al., 2014*). The difference could be that our data include cancer patients with the primary cancer of NSCLC, breast, melanoma, CRC, and kidney cancer. Also, that study was based on a single institute and a small sample size. In comparison, the median survival (10.91 months) found through this analysis was very similar to a previous study's findings for median overall survival for patients with BMs being treated with SRS (10.4 months) (*Brown et al., 2016*). A study conducted by Minniti et al. attempted to identify the neurotoxicity risks of each type of radiation treatment due to the suggestion that fractionated SRT reduces the harm of radiation on healthy tissue

**Table 3 Univariable and multivariable Cox proportional analysis of BMs patients.** Factors associated with overall survival.

| Variables | | Univariable analysis HR (95% CI) | P-value | Multivariable analysis HR (95% CI) | P-value |
|---|---|---|---|---|---|
| Age | | 1.02 (1.02–1.02) | <0.01 | 1.02 (1.01–1.02) | <0.01 |
| Treatment Combinations | SRS | 0.97 (0.91–1.03) | 0.31 | 0.94 (0.88–1.01) | 0.08 |
| | SRT | Ref | | Ref | |
| Sex | Male | Reference | | Ref | |
| | Female | 0.78 (0.74–0.83) | <0.01 | 0.82 (0.78–0.87) | <0.01 |
| Race | White | Ref | | Ref | |
| | Black | 0.95 (0.87–1.04) | 0.30 | 0.91 (0.83–0.99) | 0.03 |
| | Other | 0.72 (0.62–0.84) | <0.01 | 0.75 (0.65–0.87) | <0.01 |
| Education | >=13% NHSD | 1.08 (1.02–1.14) | 0.01 | NS | 0.21 |
| | <13% NHSD | Ref | | Ref | |
| Income | >=$35,000 | Ref | | Ref | |
| | <35,000 | 1.12 (1.06–1.18) | <0.01 | 1.11 (1.05–1.17) | <0.01 |
| Place of Living | Urban | Ref | | … | |
| | Rural | 1.12 (0.91–1.37) | 0.29 | … | |
| Hospital Type | Academic | Ref | | Ref | |
| | Community | 1.13 (1.07–1.19) | <0.01 | 1.09 (1.03–1.15) | <0.01 |
| Insurance Status | Yes | Ref | | … | |
| | No | 0.99 (0.85–1.16) | 0.92 | … | |
| Charlson/Deyo Score | 0 | Ref | | Ref | |
| | 1 | 1.22 (1.14–1.29) | <0.01 | 1.14 (1.07–1.21) | <0.01 |
| | >=2 | 1.38 (1.27–1.51) | <0.01 | 1.21 (1.11–1.32) | <0.01 |
| Chemotherapy | Yes | Ref | | | |
| | No | 1.65 (1.56–1.75) | <0.01 | 1.68 (1.58–1.79) | <0.01 |
| Primary Cancer | Breast | 0.45 (0.36–0.56) | <0.01 | 0.65 (0.51–0.82) | <0.01 |
| | NSCLC | 0.65 (0.57–0.74) | <0.01 | 0.79 (0.69–0.91) | 0.01 |
| | SCLC | 0.79 (0.65–0.96) | 0.02 | 0.97 (0.79–1.19) | 0.76 |
| | Other Lung | 0.97 (0.81–1.18) | 0.76 | 0.98 (0.80–1.19) | 0.81 |
| | Melanoma | 0.52 (0.44–0.61) | <0.01 | 0.50 (0.42–0.60) | <0.01 |
| | CRC | 0.75 (0.56–1.01) | 0.06 | 0.99 (0.73–1.34) | 0.94 |
| | Renal Cell | Ref | | Ref | |
| Year of Diagnosis | 2010–2013 | 1.10 (1.05–1.16) | <0.01 | 1.08 (1.02–1.14) | 0.01 |
| | 2014–2015 | Ref | | Ref | |

**Notes.**

Trt, treatment; NHSD, no high school degree.

(*Donovan, Parpia & Greenspoon, 2019*) and found high dose radiation treatment may be more favorable in control and reduced radiotoxicity risks for larger (>2 cm) metastatic malignancies of the brain (*Minniti et al., 2016*). Additionally, *Putz et al. (2020)* further explored the local control and risk of RN in BMs and found evidence suggesting the benefits SRT provides in treating larger metastases may also apply to BMs of smaller sizes. No significant differences in association of SRS *vs* SRT with OS when stratifying SRS and SRT by $BED_{10}$ suggests both SRS and SRT above or below $BED_{10}$ of 55 Gy included in the study are equally effective. Similarly, the study by Putz et al. did not yield a significant

**Table 4  Multivariable analyses of SRT *vs*. SRS stratified by primary cancer site.** Shows the overall survival of patients by primary cancer site.

| Tumor type | Multivariable HR (95% CI) SRT *vs*. SRS | P |
|---|---|---|
| Breast cancer | 1.14 (0.74–1.75) | 0.56 |
| NSCL | 0.94 (0.87–1.01) | 0.11 |
| SCLC | 1.31 (0.89–1.93) | 0.17 |
| Other types of lung cancer | 1.30 (0.92–1.84) | 0.14 |
| Melanoma | 0.93 (0.72–1.21) | 0.59 |
| Colon cancer | 0.42 (0.20–0.85) | 0.01 |
| Kidney cancer | 0.77 (0.54–1.09) | 0.13 |

**Notes.**

Multivariable analyses were adjusted for age at diagnosis, sex, race, health insurance status, income level, education level, place of living, comorbidity score, chemotherapy, and year of diagnosis.

difference in overall survival between patients receiving fractionated SRT compared to SRS treatment in a single dose (*Putz et al., 2020*). Interestingly, in a subset-analysis, we did find that SRT is associated with an improved OS when compared to SRS in patients with primary colon cancer. It could be that the size of BMs is usually larger in colon cancer patients than patients with other primary cancer. Due to the safety consideration, the SRS dose is usually compromised in larger tumors (*Andrews et al., 2004*). Unfortunately, the number of patients with colon cancer was too low to do the subset cox analysis after being stratified by $BED_{10}$.

To our knowledge, this study is the first to analyze the data from the NCDB comparing the characteristics and outcomes of BM patients receiving either SRS or SRT treatment. Along with the current research comparing both forms of radiation treatment, our findings warrant future studies to analyze and determine the risks presented by each form of treatment to determine if SRT could, in fact, prove to be the safer treatment plan in some cases that would currently be treated with SRS. Large scale studies with access to further details on the number of BMs, size of BMs, local recurrence, brain necrosis and cause of death would be able to provide useful information if available. However, at the very least, we corroborated that both SRS and SRT are not shown to have significant difference in overall survival with available factors considered.

One of the strengths of this study is that utilizing the large population from the NCDB allows us to better control different factors during our multivariable analysis of the association of receiving one treatment with OS. However, due to the nature of the NCDB data source, we were not able to analyze certain endpoints such as the specific cause of death, the local control of the BMs, side effects/radiotoxicity from SRS or SRT, and the control of the primary tumor. In addition, we did not have information on the size and the number of BMs, if patients had surgery on the BMs, and if the patients had extracranial metastases. Information on dose homogeneity as well as adopted techniques for treatment (gamma-knife *vs*. linac based approach) were not available. Despite these limitations, the NCDB, by representing more than 70% of newly diagnosed cancer cases nationwide, is arguably the best data source to perform a large-scale study on radiation treatment

**Table 5 Multivariable Cox proportional analysis of BMs patients with BED.** The OS of patients with different $BED_{10}$ is shown.

| Variables | | Multivariable analysis | P-value |
|---|---|---|---|
| | | HR (95% CI) | |
| Age | | 1.02 (1.01–1.02) | <0.01 |
| Biologically Effective dose ($BED_{10}$: Gy) | SRS <55 | Ref | |
| | SRS >= 55 | 0.99 (0.94–1.06) | 0.97 |
| | SRT <55 | 0.93 (0.86–1.01) | 0.07 |
| | SRT >= 55 | 1.16 (0.94–1.43) | 0.18 |
| Sex | Male | Ref | |
| | Female | 0.82 (0.77–0.86) | <0.01 |
| Race | White | Ref | |
| | Black | 0.91 (0.83–0.99) | 0.04 |
| | Other | 0.75 (0.65–0.87) | <0.01 |
| Education | >=13% NHSD | NS | 0.24 |
| | <13% NHSD | Ref | |
| Income | >=$35,000 | Ref | |
| | <35,000 | 1.11 (1.05–1.18) | <0.01 |
| Place of Living | Urban | Ref | |
| | Rural | … | |
| Hospital Type | Academic | Ref | |
| | Community | 1.08 (1.02–1.14) | 0.01 |
| Insurance Status | Yes | Ref | |
| | No | … | |
| Charlson/Deyo Score | 0 | Ref | |
| | 1 | 1.14 (1.07–1.22) | <0.01 |
| | >=2 | 1.21 (1.11–1.32) | <0.01 |
| Chemotherapy | Yes | ref | |
| | No | 1.70 (1.60–1.81) | <0.01 |
| Primary Cancer | Breast | 0.62 (0.49–0.77) | <0.01 |
| | NSCLC | 0.78 (0.68–0.90) | <0.01 |
| | SCLC | 0.98 (0.80–1.19) | 0.81 |
| | Other Lung | 0.97 (0.80–1.19) | 0.80 |
| | Melanoma | 0.46 (0.38–0.55) | <0.01 |
| | CRC | 1.00 (0.74–1.35) | 0.98 |
| | Renal Cell | Ref | |
| Year of Diagnosis | 2010–2013 | 1.10 (1.04–1.17) | <0.01 |
| | 2014–2015 | Ref | |

**Notes.**
Trt, treatment; NHSD, no high school degree.
… Variable had $p > 0.15$ in univariable analysis and was not included in the multivariable analysis.

outcomes. Our analysis provided meaningful results in a real-world setting by including a large and diverse population of patients, which are typically not achievable in clinical trials involving several clinical settings and locations.

**Table 6  Cox analysis stratified by biological effective dose and primary tumor type.** Shows overall survival of patients by primary site and $BED_{10}$.

| Tumor type | Biological effective dose | N | Multivariable HR (95% CI) | P |
|---|---|---|---|---|
| Breast cancer | SRS <55 $BED_{10}$ Gy | 38 | Ref | |
| | SRS >=55 $BED_{10}$ Gy | 89 | 0.89 (0.53–1.50) | 0.66 |
| | SRT <55 $BED_{10}$ Gy | 43 | 1.02 (0.58–1.80) | 0.95 |
| NSCL | SRS <55 $BED_{10}$ Gy | 1,579 | Ref | |
| | SRS >=55 $BED_{10}$ Gy | 2,828 | 1.01 (0.94–1.08) | 0.87 |
| | SRT <55 $BED_{10}$ Gy | 1,109 | 0.93 (0.85–1.01) | 0.09 |
| SCLC | SRS<55 $BED_{10}$ Gy | 66 | Ref | |
| | SRS>=55 $BED_{10}$ Gy | 96 | 1.16 (0.80–1.69) | 0.44 |
| | SRT<55 $BED_{10}$ Gy | 33 | 1.51 (0.92–2.46) | 0.10 |
| Other types of lung cancer | SRS<55 $BED_{10}$ Gy | 80 | Ref | |
| | SRS>=55 $BED_{10}$ Gy | 79 | 0.95 (0.66–1.37) | 0.78 |
| | SRT<55 $BED_{10}$ Gy | 59 | 1.32 (0.89–1.20) | 0.16 |
| Melanoma | SRS<55 $BED_{10}$ Gy | 130 | Ref | |
| | SRS>=55 $BED_{10}$ Gy | 190 | 0.82 (0.62–1.09) | 0.18 |
| | SRT<55 $BED_{10}$ Gy | 120 | 0.83 (0.61–1.13) | 0.24 |
| Kidney cancer | SRS<55 $BED_{10}$ Gy | 70 | Ref | |
| | SRS>=55 $BED_{10}$ Gy | 133 | 0.96 (0.70–1.33) | 0.82 |
| | SRT<55 $BED_{10}$ Gy | 42 | 0.67 (0.43–1.04) | 0.07 |

**Notes.**
The number of patients with colon cancer is too low to do the subset cox analysis.

# CONCLUSIONS

This study found that receiving either SRT or SRS was not significantly associated with a difference in OS. With these findings, further studies are warranted to determine if SRT is less toxic than SRS either overall, or in specific cases.

## Funding

This work is supported by the research grant from Otis Glebe Medical Foundation Nebraska University Foundation #01100674. The funders had no role in study design, data collection and analysis, decision to publish, or preparation of the manuscript.

## Grant Disclosures

The following grant information was disclosed by the authors:
The research grant from Otis Glebe Medical Foundation Nebraska University Foundation: #01100674.

## Competing Interests

The authors declare there are no competing interests.

## Author Contributions

- Garett Paul Ostdiek-Wille conceived and designed the experiments, performed the experiments, prepared figures and/or tables, authored or reviewed drafts of the article, and approved the final draft.
- Saber Amin conceived and designed the experiments, performed the experiments, analyzed the data, prepared figures and/or tables, authored or reviewed drafts of the article, and approved the final draft.
- Shuo Wang conceived and designed the experiments, performed the experiments, authored or reviewed drafts of the article, and approved the final draft.
- Chi Zhang conceived and designed the experiments, performed the experiments, authored or reviewed drafts of the article, and approved the final draft.
- Chi Lin conceived and designed the experiments, performed the experiments, prepared figures and/or tables, authored or reviewed drafts of the article, and approved the final draft.

## Human Ethics

The following information was supplied relating to ethical approvals (i.e., approving body and any reference numbers):

De-identified data was used from the NCDB, which does not require an IRB.

## Data Availability

The data is available at the National Cancer Database: https://www.facs.org/quality-programs/cancer-programs/national-cancer-database/.

To obtain access to the NCDB data, your institution must be a participating member of the NCDB. Researchers must create an account and then submit application for data access, which includes the summary of the hypothesis, plausibility of the research questions, and statistical analyses.

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
