# Peer review of "Single fraction stereotactic radiosurgery and fractionated stereotactic radiotherapy provide equal prognosis with overall survival in patients with brain metastases at diagnosis without surgery at primary site"

_PeerJ, doi:10.7717/peerj.15357_

## Round 0.1 · original submission · Major Revisions

Dear Dr. Amin,

Thank you for submitting your manuscript "Single fraction stereotactic radiosurgery and fractionated stereotactic radiotherapy provide equal prognosis with overall survival in patients with brain metastases at diagnosis without surgery at primary site" to PeerJ. We have now received reports from the reviewers, and, after careful consideration internally, we have decided to invite a major revision of the manuscript.

As you will see from the reports copied below, the reviewers raise important concerns regarding the rationale design of the study and supporting conclusion of SRT over SRS. We find that these concerns limit the strength of the study, and therefore we ask you to address them with additional work. Without substantial revisions, we will be unlikely to send the paper back for review.

If you feel that you are able to comprehensively address the reviewers’ concerns, please provide a point-by-point response to these comments along with your revision. Please show all changes in the manuscript text file with track changes or color highlighting. If you are unable to address specific reviewer requests or find any points invalid, please explain why in the point-by-point response.

Thanks

Abhishek Tyagi, PhD
Academic Editor,
PeerJ

·

Basic reporting

The topic is intriguing.

The text is clear, a little bit essential but adequate to the context.

Experimental design

The number of analyzed cases is high and the analysis of the data seems to be conducted correctly.

Validity of the findings

The number of metastatic lesions as a predictor/variable has not been reported or evaluated. It could be a reason for significant criticism as well as the median volume of the treated lesions(even if declared in the text).

Additional comments

I recommend as followings:
review the acronyms. Most of them are note explained : (I.e: BM, CRC, etc...!)

Reviewer 2 ·

Basic reporting

.

Experimental design

.

Validity of the findings

.

Additional comments

Abstract
Background
-Line 25: is it unknown? There is no randomised comparison but there are meta-analyses/trials
-Line 26: why was OS chosen? Wouldn’t local control/RN etc be a better way to evaluate effectiveness and safety

Results
-Line 33: suggest round to single decimal for percent & median OS + round to 2 decimals for p value
-Line 34: 2nd/3rd sentence = saying the same thing as final sentence?\

Introduction
-Line 53: suggest expansion to clarify reason for difference in OS bw histologies as it is not generally due to intracranial disease
-Line 59: ?Ref 9. Met >4cm would be likely for surgery, many BMs= SRS/SRT
-Line 79: reference for 25% risk RN? This seems too high
-Suggest make introduction more concise. E.g. 69-75 is somewhat well known knowledge

Line 104: no unit of measurement for dose
Line 118: “The receipt of SRS or SRT is the main variable of OS”. Do you mean that the primary factor in determining OS is whether or not a patient underwent SRS or SRT?

Line 203: Why BED 55 Gy?
Line 214/216: Can you expand on potential reasons why your OS data is slightly different?

Line 233: I can find multiple previous papers which have analysed NCDB data for SRS and brain metastases
Line 235: As your findings have found no difference, which aspect of your findings warrants future studies?

Reviewer 3 ·

Basic reporting

I enjoyed reading the manuscript “Single fraction stereotactic radiosurgery and fractionated stereotactic radiotherapy provide equal prognosis with overall survival in patients with brain metastases at diagnosis without surgery at primary site.”

The manuscript reads well and is clearly structured. The tables and figures are well designed, and the statistical analysis is done and outlined correctly. However, the rationale for the study design and the research question asked by the authors needs to be reviewed.
While it is true that no prior studies have compared the outcomes of fractionated SRT with single fraction SRS in terms of prognosis and overall survival in patients with brain metastases, I must question to what extent this research question is relevant to this specific cancer population.
A priori it is not expected for the number of fractions with similar BED to have an impact on OS in such heterogenous population, where many other factors play a role in determining the outcome (death). As it has been shown by multiple studies in the past, both regimens (one vs multiple fractions) are comparable schemes in terms of BMs local control and toxicity profile. The decision on choosing single or multiple fraction RT for intact BMs depends to a lower extent by the predictors analyzed by the authors, and more importantly on the characteristics of intracranial disease, i.e. on the location and size of the lesion, number of BMs at diagnosis, status of extracranial disease, GPA or RPA scores. The outcome in terms of OS in this population of patients is strongly determined by distant brain failure and intracranial progression free survival rather than the rate of local failure or toxicity following SRS vs. SRT. It is predictable that the authors did not see any statistically significant differences between both groups.
In addition, it is not reported whether the event of death, used to calculate OS, was a CNS-related death or death from other causes (not cancer related, extracranial disease progression, fatal treatment-related toxicity…).
Another significant downside is the lack of analysis of BMs associated factors, that are important as mentioned, when deciding between SRS or SRT treatment and also crucial for prognosis. These include, among others, size/volume and location of the BMs treated, presence of BMs associated symptoms, as well as the number of BMs at diagnosis, which has been shown to be associated with the intracranial progression free survival. The data on presence or absence of leptomeningeal disease or prior WBRT are also lacking and are crucial in determining the prognosis in patients with newly diagnosed BMs.

The authors should review the purpose of analyzing OS as primary endpoint in this study with the lacking data on RT modality and characteristics of the treated BMs. It would be more meaningful to analyze the patterns of toxicity and local control with each RT modality and analyze in more depth the information on BMs and RT, if possible, to obtain at the cost of including a lower number of participants.
Due to the design of this study, it seems inappropriate to draw a conclusion that SRT could be safer that SRS.

Minor comments:

Line 60: “…or a metastasis greater than four centimeters in size would all likely receive WBRT”. This statement is disputable. Sizeable lesions can be managed with local brain-directed therapies (surgery + RT to surgical bed, fractionated RT only..)
Line 70: “SRS… is associate with some radiation toxicity”. I would suggest being more specific and applying the term of radionecrosis. Another toxicity could consist of intralesional hemorrhage, though data on its risk following SRS are limited.

Line 72: “…could potentially help specific areas of the brain”- it would be helpful if authors explain what is meant by that statement.
Line 79, 83: Consistency in using the abbreviation of “RN” is needed
Line 84: unsure why RN would translate into OS benefit, maybe the authors could consider explaining that statement.
Line 107: Consistency in using the abbreviation of “BMs” is needed
Lines 105: Please include the BED for each fractionation scheme

Experimental design

.

Validity of the findings

.

Additional comments

.

---

## Round 0.2 · accepted · Accept

Dear Dr. Amin,
Thank you for your submission to PeerJ.
I am writing to inform you that your manuscript - Single fraction stereotactic radiosurgery and fractionated stereotactic radiotherapy provide equal prognosis with overall survival in patients with brain metastases at diagnosis without surgery at primary site - has been Accepted for publication.

Congratulations!

With kind regards,
Abhishek Tyagi
Academic Editor
PeerJ Life & Environment